# A Cross-Sectional Retrospective Study of Non-Splenectomized and Never-Treated Patients with Type 1 Gaucher Disease

**DOI:** 10.3390/jcm9082343

**Published:** 2020-07-22

**Authors:** Christine Serratrice, Jérôme Stirnemann, Amina Berrahal, Nadia Belmatoug, Fabrice Camou, Catherine Caillaud, Thierry Billette de Villemeur, Florence Dalbies, Bérengère Cador, Roseline Froissart, Agathe Masseau, Anaïs Brassier, Bénédicte Hivert, Laure Swiader, Ivan Bertchansky, Claire de Moreuil, Brigitte Chabrol, Isabelle Durieu, Vanessa Leguy Seguin, Leonardo Astudillo, Sébastien Humbert, Samia Pichard, Catherine Marcel, Isabelle Hau Rainsard, Monia Bengherbia, Karima Yousfi, Marc G. Berger

**Affiliations:** 1Department of Internal Medicine for the Aged, Geneva University Hospitals, 1226 Thonex- Geneva, Switzerland; 2Department of Internal Medicine, Department of Medicine, Geneva University Hospitals, 1205 Geneva, Switzerland; Jerome.stirnemann@hcuge.ch; 3University Hospital of Clermont Ferrand, Hematology Biology Department, 63000 Clermont-Ferrand, France; aberrahal@chu-clermontferrand.fr (A.B.); mberger@chu-clermontferrand.fr (M.G.B.); 4Department of Internal Medicine, Centre de Référence des Maladies Lysosomales, AP-HP.Nord, Site Beaujon, Paris University, 92110 Clichy, France; nadia.belmatoug@aphp.fr (N.B.); marcel.catherine@orange.fr (C.M.); monia.bengherbia@aphp.fr (M.B.); karima.yousfi@aphp.fr (K.Y.); 5Intensive Care Unit, Hôpital Saint-André, CHU Bordeaux, 33000 Bordeaux, France; fabrice.camou@chu-bordeaux.fr; 6Biochemistry, Metabolomics, and Proteomics Department, Necker Enfants Malades University Hospital, AP-HP. Center-Paris University, 75015 Paris, France; catherine.caillaud@inserm.fr; 7Department of Pediatric Neurology, Hôpital Trousseau, AP-HP, 75012 Paris, France; thierry.billette@aphp.fr; 8Institute of Cancerology and Hematology, CHRU Morvan, 29200 Brest, France; florence.dalbies@chu-brest.fr; 9Department of Internal Medicine, CHU Pontchaillou, 35000 Rennes, France; berengere.cador@chu-rennes.fr; 10Biochemical and Molecular Biology Department, Centre de Biologie et de Pathologie Est, Hospices Civils de Lyon, 69500 Bron, France; roseline.froissart@chu-lyon.fr; 11Department of Internal Medicine, CHU Hôtel Dieu, 44093 Nantes, France; agathe.masseau@chu-nantes.fr; 12Reference Centre for Hereditary Metabolic Diseases, Hôpital Necker-Enfants Malades, AP-HP, IHU Institut Imagine, 75015 Paris, France; anais.brassier@aphp.fr; 13Department of Clinical Hematology, Hôpital Saint Vincent de Paul, Groupement des Hôpitaux de l’Institut Catholique de Lille, 59800 Lille, France; Hivert.Benedicte@ghicl.net; 14Department of Internal Medicine, CHU Timone, 13005 Marseille, France; laure.swiader@ap-hm.fr; 15Department of Internal Medicine, Saint-Eloi Hospital, CHU Montpellier, 34000 Montpellier, France; i-bertchansky@chu-montpellier.fr; 16Department of Internal Medicine, Hôpital de la Cavale Blanche, CHRU de Brest, 29200 Brest, France; claire.demoreuil@chu-brest.fr; 17Department of Pediatric Neurology, CHU Timone, 13005 Marseille, France; brigitte.chabrol@ap-hm.fr; 18Department of Internal Medicine, Hospices Civils de Lyon, University Claude Bernard Lyon 1, 69495 Pierre-Bénite, France; isabelle.durieu@chu-lyon.fr; 19Department of Internal Medicine and Clinical Immunology, CHU F. Mitterrand, 21000 Dijon, France; vanessa.leguy.seguin@chu-dijon.fr; 20Department of Internal Medicine, CHU Toulouse, 31300 Toulouse, France; astudillo.l@chu-toulouse.fr; 21Department of Internal Medicine, Besancon University Hospital, 25000 Besancon, France; shumbert@chu-besancon.fr; 22Service of Metabolic Diseases, Hôpital Robert Debré, 75019 Paris, France; samia.pichard@aphp.fr; 23Service of General Pediatry, CHI Créteil, 94000 Créteil, France; isabelle.hau@chicreteil.fr; 24University Clermont Auvergne, EA 7453 CHELTER, 63000 Clermont-Ferrand, France

**Keywords:** Gaucher disease, lysosomal storage disorder, disease course, enzyme replacement therapy, risk factors, prognosis

## Abstract

Patients with type 1 Gaucher disease (GD1) present thrombocytopenia, anemia, organomegaly, and bone complications. Most experts consider that the less aggressive forms do not require specific treatment. However, little is known about the disease course of these forms. The objective of this cross-sectional retrospective study was to compare the clinical, radiological, and laboratory characteristics of patients with less severe GD1 at diagnosis and at the last evaluation to identify features that might lead to potential complications. Non-splenectomized and never-treated patients (19 women and 17 men) were identified in the French Gaucher Disease Registry (FGDR). Their median age was 36.6 years (2.4–75.1), and their median follow-up was 7.8 years (0.4–32.4). Moreover, 38.7% were heterozygous for the *GBA1* N370S variant, and 22.6% for the *GBA1* L444P variant. From diagnosis to the last evaluation, GD1 did not worsen in 75% of these patients. Some parameters improved (fatigue and hemoglobin concentration), whereas platelet count and chitotriosidase level remained stable. In one patient (2.7%), Lewy body dementia was diagnosed at 46 years of age. Bone lesion onset was late and usually a single event in most patients. This analysis highlights the genotypic heterogeneity of this subgroup, in which disease could remain stable and even improve spontaneously. It also draws attention to the possible risk of Lewy body disease and late onset of bone complications, even if isolated, to be confirmed in larger series and with longer follow-up.

## 1. Introduction

Gaucher disease (GD) is a rare lysosomal autosomal recessive disease caused by deficiency of the lysosomal enzyme acid beta-glucosidase or glucocerebrosidase [1], or in rare cases of its activator saposin C [2]. Its prevalence is about 1/60,000 worldwide, but 1/6,000 in Israel. In the French Gaucher Disease Registry (FGDR), prevalence is estimated at 1/140,000 [3]. GD diagnosis is confirmed by the demonstration of low glucocerebrosidase activity usually measured in peripheral blood leukocytes [4]. In most cases, GD shows significant morbidity and it is classified into three phenotypes. Type 1 Gaucher disease (GD1) is characterized by a combination of splenomegaly and/or hepatomegaly, cytopenia (thrombocytopenia, anemia, and more rarely leukopenia) of varying degrees, and/or bone disease (Erlenmeyer flask deformity, bone infiltration, osteoporosis, lytic lesions, pathologic and vertebral fractures, bone infarction, and avascular necrosis leading to degenerative arthropathy). GD1 represents 90% of all cases. Type 2 GD is an acute lethal neuronopathic form, and type 3 GD is an intermediate form (patients who survived infancy but have some forms of neurologic involvement). Within each GD type, phenotypic heterogeneity is important and poorly understood [5].

Among more than 450 *GBA1* gene variants described, only a limited number appears to be related to a specific GD phenotype. For instance, the c.1448T > C or p.Leu483Pro (also called L444P) variant predisposes to neurological involvement when present in a homozygous state, whereas the c.1226A > G or p.Asn409Ser (N370S) variant (either heterozygous or homozygous) is associated with the non-neuronopathic form. However, for other variants, genotype–phenotype correlation data are not sufficiently robust to predict the disease aggressiveness.

The gold standard treatment is enzyme replacement therapy (ERT) (imiglucerase (CEREZYME^®^ Sanofi Genzyme, Cambrige, USA), velaglucerase alpha (VPRIV^®^ Takeda Shire, Tokyo, Japan), taliglucerase (ELELYSO^®^ Pfizer, New York, USA)) that is effective for most GD symptoms. Two substrate reducers are also available, namely miglustat (Zavesca) (Actelion Ltd., Basel, Switzerland.) and eliglustat (Cerdelga) (Sanofi Genzyme, Cambridge, USA). Due to its low to medium efficacy, and the important neurologic and gastrointestinal side effects, miglustat has been approved only as second-line therapy for patients unsuitable for ERT. Eliglustat was approved as first-line therapy for adults with GD1 in the USA and Europe in 2014 and 2015, respectively.

ERT has dramatically changed GD1 prognosis, leading to an increase in the duration and quality of life. The long-term follow-up of such patients has revealed unsuspected consequences of the enzyme deficiency, such as higher frequency of cancer [6] and Parkinson’s disease [7]. However, the indication for treatment in patients with milder forms remains controversial; most experts agree that not all patients need to be specifically treated [8,9]. Many countries have their own criteria to start treatment. In France, treatment indication is determined by a multidisciplinary working group when a certain number of criteria are met [10]. However, as some untreated patients could be at risk of developing severe complications (e.g., bone involvement, hemorrhages, Parkinson’s disease), increased surveillance might be desirable. It is also important to gather more knowledge on the long-term disease course in never-treated patients with less aggressive GD, to better adapt their management. However, very few and often contradictory data are available in the literature on the natural course of untreated patients with GD [9,11,12,13,14,15,16,17,18,19]. Most studies are retrospective, with heterogeneous cohorts and fragmented data, and have also included splenectomized patients. Nevertheless, they suggest that some patients could remain untreated without disease progression. The most recent and important study [9] concerned a genotypically homogeneous cohort. However, in France, analysis of the FGDR data highlighted the important genotypic and phenotypic heterogeneity of patients with GD1 [3]. Therefore, we started a study on untreated patients with GD1 in France to retrospectively (this analysis) and prospectively collect information on the clinical characteristics of these patients and on the natural course of GD1, particularly of the less severe forms.

## 2. Patients and Methods

This multicentric French national study is an observational retrospective cohort study based on the cross-sectional analysis of data from the FGDR [3]. The study was approved by the Sud Est VI Ethics Committee. All patients signed a consent form to be included in the FGDR. The study focused on the 147 patients registered in the FGDR as untreated (i.e., no splenectomy, ERT, or substrate inhibitor). All collected data were systematically verified.

### 2.1. The French Gaucher Disease Registry (FGDR)

The FGDR was developed in 2009 by the Referral Center for Lysosomal Diseases and its Committee of Evaluation of GD Treatment. Its objectives were to improve GD management and professional practices and to collect epidemiological data. It includes all patients with GD followed in France since 1980. Clinical information as well as laboratory and bone data at GD diagnosis and throughout the follow-up were recorded, with the identification of intercurrent events, particularly bone complications and malignancies. It was approved by the French Data Protection Commission and is certified by the French Institute for Public Health Surveillance and the National Institute of Health and Medical Research (INSERM) [3].

### 2.2. Patients’ Selection

The first screening of the FGDR data allowed eliminating patients who no longer live in France and patients lost to follow-up. Then, the records of all patients listed as untreated in the FGDR were screened. The second step was to contact the physician in charge of each patient to ensure that patients were still alive, followed, and untreated. As splenectomy can be considered a specific treatment, splenectomized patients were excluded from the study. If the physician and patient both accepted to participate in the study, the patient signed an informed consent form, and then additional data were collected to complete the FGDR information. The FGDR database was last screened on 3 June 2019.

### 2.3. Statistical Analyses

The characteristics of never-treated patients with GD1 were described using medians (interquartile range) for continuous data and numbers (%) for categorical data. The Student’s *t*-test or Mann–Whitney–Wilcoxon test were used for comparing quantitative data, and the chi-squared test or Fischer test for categorical data, as appropriate. A two-sided *p* < 0.05 was considered significant. Time to first bone symptoms was estimated with the Kaplan–Meier method because only the first bone symptoms were considered. Data were censored when no bone symptom occurred before data freezing. Changes over time of the main GD-linked parameters (platelets, hemoglobin, leucocytes, Chemokine C-C motif ligand 18 (CCL18), chitotriosidase, ferritin) were analyzed using linear mixed models for repeated measures. This model is a generalization of a standard linear regression analysis that allows modeling the parameter changes for each individual over time and takes into account the intra-individual variations. The slopes for each parameter were identified and variation was tested. Correlations between continuous variables were tested with the non-parametric Spearman’s correlation method. Hemoglobin was defined as the percentage of the normal value (x N), according to the World Health Organization (WHO) reference values [20]. Statistical analyses were performed using the R statistical software package, version 3.1.1.

### 2.4. Data Analyzed

For each parameter, only the patients for whom data were available in the FGDR at diagnosis (or +1 year) and at the last evaluation were considered. This was a retrospective study, and consequently some data were missing, particularly during the diagnosis phase. As clinical and laboratory parameters were relatively stable in untreated patients after GD diagnosis, data for the first year after diagnosis were considered similar to those at diagnosis. Splenomegaly and hepatomegaly (presence/absence) were considered at diagnosis and at the last evaluation. When specific measures were available for different time points, the oldest and the most recent were considered.

## 3. Results

In the FGDR, 658 patients were registered, among whom 467 were still alive. Of these, 320 patients were treated and 147 untreated. In total, 36 patients could be included in the present analysis (Figure 1). The patients’ characteristics are described in Table 1.

The changes in the main clinical and biological GD1 parameters between diagnosis and last evaluations are summarized in Table 2. All patients with asthenia at diagnosis did not report this symptom at the last follow-up. Conversely, two patients had asthenia only at the last evaluation. Chronic bone pain disappeared spontaneously in the patient who had it at diagnosis, and was reported by five patients at the last evaluation. One patient (a 9-year-old girl) had a bone crisis at the last evaluation (5 years after diagnosis). In this girl, GD1 was diagnosed at the age of 4 because of isolated splenomegaly, without thrombocytopenia (platelet count: 189 × 10⁹/L). Her *GBA1* genotype was c.1226A > G/c.1448T > C (N370S/L444P). Bone imaging did not show any fracture, osteonecrosis, or bone infarct.

Concerning hepatosplenomegaly, patients with missing data for spleen and/or liver were excluded from the analysis. Spleen size (i.e., length) increased in nine patients, spontaneously decreased in one patient, and remained stable in five patients. The median follow-up was 10 years (min–max: 3–31). At the last evaluation, hepatomegaly was not recorded in two patients who had it at diagnosis and was detected in four patients without hepatomegaly at diagnosis. Analysis of the available data indicated that hepatomegaly was mild (axis length between 15 and 16.8 cm).

Between diagnosis and last evaluation, only hemoglobin concentration significantly increased (*p* < 0.001). Conversely, platelet and leucocyte counts as well as chitotriosidase, ferritin, and CCL18 levels did not significantly change. According to the WHO criteria based on age and sex, eight patients (six men and two women, including three children) had anemia at diagnosis (median: 0.94 × *N* (min 0.79 × *N*- max 0.98 × *N*)), and seven patients at the last evaluation (four men and three women, including three children). Anemia was corrected in four patients who had it at diagnosis, while hemoglobin concentration remained stable in three patients, and decreased in one patient during the follow-up. In three patients, anemia appeared during the follow-up.

Platelet counts spontaneously increased by more than 25% in two patients and decreased by more than 25% in eight patients. In the other patients (*n* = 26), platelet count remained stable (change <25%), according to the criteria for patients with GD1 receiving maintenance treatment with imiglucerase [21]. The median glucosylsphingosine (lyso-Gb1) level at the last evaluation was 113 (7.86–259) ng/mL and was negatively correlated with platelet count (*p* = 0.03) (Appendix A).

Among the 36 patients, some met the criteria for treatment initiation at diagnosis or at last evaluation (Table 3). At diagnosis, two patients should have been treated, according to the French treatment criteria revised in 2015 [10], due to the presence of bone complications. Both patients refused treatment, and GD1 parameters did not worsen (two and seven years of follow-up, respectively). At the last evaluation, eight patients met the treatment criteria, including the two patients with bone manifestations at diagnosis. In the other six patients, symptoms worsened during the follow-up. The *GBA1* genotype was available for 7/8 patients. They all were heterozygous for the N370S variant and 3/8 carried also the L444P variant (compared with 4 of the 23 patients without treatment criteria; *p* = 0.33). The main reason for not starting treatment was patient refusal.

In 15 patients (42%), GD1 was diagnosed before the age of 18 years (Table 4). Ten were still younger than 18 years old at the last evaluation. Three met the criteria for treatment initiation during the follow-up and two were still children when GD1 worsened. In one patient (GD1 diagnosis at the age of 3 years because of splenomegaly), platelet count was 50 × 10⁹/L at the age of 9, and the reason of no treatment was unknown. One young patient had chronic bone pain and one bone crisis, but no bone infarction, osteonecrosis, or bone fracture. Bone MRI showed only medullar infiltration. One patient (platelet count of 86 × 10⁹/L and hemorrhagic syndrome) refused treatment.

Lewy body dementia was diagnosed in a 46-year-old patient (2.7% of 36). This prevalence is similar to what is found in the FGDR (2.4%), but much higher than in the general population in France (0.29%) [22]. Monoclonal gammopathy was detected in two patients (5.5%), but no myeloma or lymphoma. Seven women had 11 pregnancies with 9 live births. Moreover, one spontaneous abortion and one medical abortion were reported. The diagnosis of GD was made after pregnancy in four patients.

## 4. Discussion

GD1 was the first lysosomal storage disorder to be treated by ERT. Although the impact on life expectancy remains difficult to demonstrate due to the lack of an equivalent comparative group of untreated patients, ERT has dramatically modified the patients’ outcomes [18,23,24]. For some patients with poorly symptomatic disease, ERT benefit might appear negligible relative to its cost, and they are often not treated. However, it is very difficult to predict which patients will worsen and to evaluate the risk of not treating. Therefore, better understanding the natural disease course in this subgroup of patients with GD1 is essential. Previous studies analyzed the natural course of untreated patients with GD1 [9,11,12,13,14,16,17,18,19]. All these studies included splenectomized patients. Dinur et al. described the largest cohort of untreated patients (*n* = 103), but 8.7% of them were splenectomized [9]. Splenectomy can influence GD1 natural course, and particularly platelets count. Consequently, it seemed important to exclude splenectomized patients from the analyses on untreated patients with GD1. In the cohort of the study by Dinur et al., 80.5% of patients carried the N370S/N370S mutation that is usually associated with a less severe disease phenotype. However, this genotype is not common in other countries. Conversely, our cohort of never-treated and non-splenectomized patients presented an important genotypic heterogeneity. Indeed, about 22% of these patients carried the L444P variant that is usually related to a more severe form; however, half of them did not meet the treatment indication criteria. This confirms the lack of robustness of the genotype–phenotype relationship as already observed by Balwani et al. [25], and suggests the contribution of other genetic, epigenetic, and environmental factors.

Our study focused on the clinical, biological, and radiological changes over time (from diagnosis to the most recent evaluation). In most patients (87%), GD1 was diagnosed after 1991, when the first ERT became available in France. Therefore, the choice of not treating was not related to the absence of treatment options, but on the physician’s and mainly the patient’s decision, which might not be related just to disease severity.

Our temporal analysis shows that asthenia, which is a very common symptom in GD1, can spontaneously improve. Fatigue is certainly one of the most difficult symptoms to analyze, particularly in patients with GD1 [26], also due to the absence of a validated specific tool. Fatigue in GD1 is not exclusively explained by hematological alterations, such as anemia, and other causes should be explored [27]. Overall, laboratory parameters did not change over time, except hemoglobin concentration that improved, differently from what was reported by Chi and Amato [18]. The stability of platelet count and GD1 biomarkers was previously observed [12,15,18]. Moreover, we confirmed in a more genotypically heterogeneous population the negative correlation between glucosylsphingosine level and platelet count previously reported by Dinur et al. Eight patients (22%) filled the criteria for treatment initiation at the last evaluation, and two of them (5.5%) already at diagnosis. In six of them, treatment was indicated due to bone involvement (pathological fractures, osteonecrosis, or bone infarction). Although the appearance of a single bone event is considered a sufficient criterion for treatment initiation by the French expert committee in order to prevent other bone complications, the existence of never-treated patients with only one bone complication during a long follow-up raises the question of the relevance of this criterion. However, we currently cannot distinguish patients who will have a single bone complication from those who will present other bone events. Large-scale studies are needed to identify biomarkers for the early detection of patients at risk of bone lesions. The prevalence of bone events in our cohort was lower than in other studies that also included patients with splenectomy, which may have increased the risk [12,16]. Thrombocytopenia with or without hemorrhagic syndrome justified treatment initiation in the other two patients. This is rare and in line with the results by Dinur et al. [9]. Most experts recommend treating all symptomatic pediatric patients [28]. In this study, only 10 of the 72 children included in the FGDR were never treated. It would be important to obtain precise data on the natural history of GD1 in children. Beutler [12] and Zimran [11] proposed that GD1 progression usually occurs during childhood and less often during adulthood. Conversely, Maaswinkel-Mooij [14] found that GD1 can worsen at any age in non-Jewish patients. This discrepancy could be partly related to genotypic differences between Israeli and Dutch patients. In terms of long-term comorbidities, the prevalence of Parkinson’s disease and Lewy body dementia in never-treated and non-splenectomized patients was similar to that of the whole FGDR, suggesting that patients with less aggressive phenotypic forms may also be at higher risk of Parkinson’s disease compared with the general population. This is in agreement with the influence of *GBA* mutations (even in carriers) on Parkinson’s disease onset and symptomatology [29]. Moreover, as most patients with GD1 do not develop Parkinson’s disease, with a difference even among siblings with GD [30], a larger cohort with a long-term follow-up of all patients is needed to confirm this finding and to determine the underlying factors.

Our study has some limitations as follows: (1) we did not have volume measurement for visceral involvement, but only organ length that may be subject to inter- and intra-individual variations; (2) we could not include all never-treated patients in France, particularly those lost to follow-up who may be the patients with the least severe manifestations; (3) we could not systematically assess changes in glucosylsphingosine level between diagnosis and last evaluation because the dosage at diagnosis was not available for most patients; and (4) the sample size remains limited.

## 5. Conclusions

Our study describes the natural changes in the main parameters in never-treated and non-splenectomized patients with GD1 from diagnosis to last evaluation and confirms that some symptoms (e.g., asthenia) can spontaneously improve. In this cohort of limited size, *GBA*1 genotype did not seem to be correlated with the disease course. Therefore, long-term surveillance is also required in patients with less severe disease. Unfortunately, we could not identify specific profile(s) that might predict disease worsening in this group of patients. The prospective follow-up of this cohort will certainly bring some insights into this crucial issue.

## Figures and Tables

**Figure 1 jcm-09-02343-f001:**
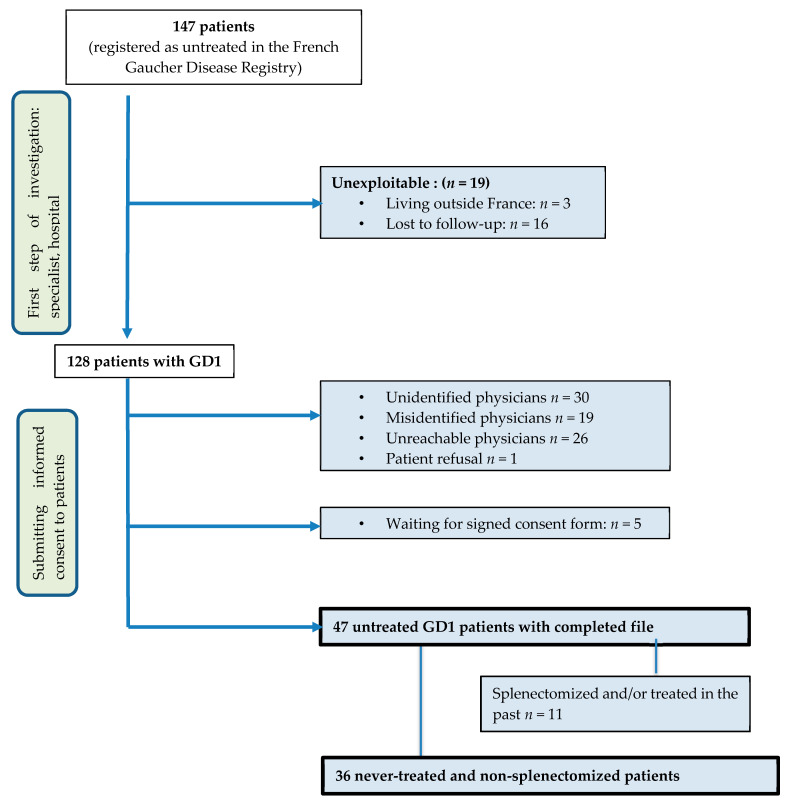
Patients’ inclusion flowchart. GD1: type 1 Gaucher disease.

**Table 1 jcm-09-02343-t001:** Patients’ characteristics.

	Never-Treated and Non-Splenectomized Patients with GD1, *n* = 36
Median age at first symptoms	17.5 (0–59)
Median age at diagnosis	24.8 (0.1–72.1)
Median age at the last evaluation	36.6 (2.4–75.1)
Median time between diagnosis and last evaluation (years)	7.8 (0.4–32.4)
Sex ratio F/M	19/17
*GBA1* genotype N370S/N370S (%) N370S/other (non L444P) (%) L444P/N370S (%) L444P/other (%) Other/Other (%)	*n* = 3 17/31 (22.6) 12/31 (38.7) 5/31 (16) 2/31 (6) 5/31 (16.1)
Conditions leading to diagnosis (several possibilities) Splenomegaly (%) Thrombocytopenia (%) Family screening (%) Unknown (%)	47 42 16 14

GD1: type 1 Gaucher disease; F/M: female/male.

**Table 2 jcm-09-02343-t002:** Changes in clinical and biological parameters between diagnosis and last evaluation.

Clinical Parameters	Diagnosis	Last Evaluation	Biological Parameters	Slope	*p*-Value
Asthenia	6/34 (17.6%)	2/34 (6%)	Hemoglobin, g/dL/year	+0.05	<0.001
Abdominal pain	0/34 (0%)	2/34 (6%)	Platelets, giga/L/year	−1456.57	0.075
Chronic bone pain	1/34 (3%)	5/34 (14.7%)	Leucocytes, giga/L/year	−24.71	0.487
Acute bone crisis	0/34 (0%)	1/34 (3%)	Chitotriosidase, nmol/h/mL/year	−66.76	0.456
Splenomegaly	19/28 (67.8%)	20/28 (71.4%)	Ferritin, ng/mL/year	+1.66	0.807
Hepatomegaly	10/16 (62.5%)	12/16 (75%)	CCL18, ng/mL/year	−6.81	0.182

**Table 3 jcm-09-02343-t003:** Patients with criteria for treatment initiation * at diagnosis and at the last evaluation.

	*GBA1* Genotype Amino Acid Change (Nucleotide Change)	Age at Diagnosis	Age at Last Evaluation	Age at Treatment Criteria Fulfillment	Criteria for Treatment at Diagnosis	Criteria for Treatmentat Last Evaluation	Reason for Absence of Treatment
M	N370S/V121A (c.1226A > G/c.479T > C)	72	74	Not known	Bilateral aseptic osteonecrosis **	Bilateral aseptic osteonecrosis	Patient’s refusal
M	N370S/L444P (c.1226A > G/c.1448T > C)	53	60	43	Humeral fracture Bilateral femoral infarcts	Humeral fracture Bilateral femoral infarcts	Patient’s refusal
M	N370S/S125N (c.1226A > G/c.491G > A)	4	9	9	-	Thrombocytopenia 50 × 10⁹/L	Not known
M	ND/ND	17	47	40	-	Thrombocytopenia (86 × 10⁹/L) + hemorrhagic syndrome	Patient’s refusal
F	N370S/L444P (c.1226A > G/c.1448T > C)	4	9	9	-	Bone crisis	Treated 1 year after inclusion
F	N370S/L444P (c.1226A > G/c.1448T > C)	61	63	63	-	Vertebral fracture L5	Severe Lewy body dementia
F	N370S/ND (c.1226A > G/ND)	32	55	45	-	Femoral infarction	Patient’s refusal
M	N370S/N370S (c.1226A > G/c.1226A > G)	51	69	69	-	Vertebral fracture	Patient’s refusal

* According to the French National Diagnosis and Treatment Protocol. ** Diagnosis of osteonecrosis performed during the routine visit for GD1 diagnosis. ND: not determined.

**Table 4 jcm-09-02343-t004:** Children’s characteristics.

	***n* = 15**
Median age at diagnosis (years)	4.2 (0.1–17)
Median age at last follow-up (years)	8.6 (2.4–17.6)
Median follow-up duration (years)	4.4 (2.1–15.5)
Sex ratio F/M	7/8
*GBA1* genotype N370S/N370S (%) N370S/other (non L444P) (%) N370S/L444P (%) Other	***n* = 10** 0/10 (0) 6/10 (60) 2/10 (20) 2/10 (20)

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
