# Peer review of "A Cross-Sectional Retrospective Study of Non-Splenectomized and Never-Treated Patients with Type 1 Gaucher Disease"

_jcm, 2020, doi:10.3390/jcm9082343_

Round 1

Reviewer 1 Report

The authors present a register study on untreated patients with GD1 in France. The important issues points of the study are the clinical course of the disease without therapeutic interventions, phenotypic-genotypic correlations in the French cohort as well as the importance of keeping detailed and updated registries of the disease.

The drawback of the study is a major drop out of patients on whom detailed data were available, from initial 147 to final 36 patients. Given that, the manuscript in the current form is too long, and should be presented in a shortened form, with fewer and more detailed tables, omitting descriptive part concerning these data in the text, and shortening the discussion. Data concerning children can still be presented seperately from adults.

Author Response

Reviewer #1:

The drawback of the study is a major drop out of patients on whom detailed data were available, from initial 147 to final 36 patients. Given that, the manuscript in the current form is too long, and should be presented in a shortened form, with fewer and more detailed tables, omitting descriptive part concerning these data in the text, and shortening the discussion. Data concerning children can still be presented separately from adults.

The manuscript has been shortened according to the comment; particularly, the patients’ description has been removed from the text and added to Table 1. Table 2 and Table 3 have been combined in one table as well as Table 4 and Table 5. The data and Table about children are still separated, as suggested by the reviewer.

The Results section has been reduced (1155 words including tables versus 1682 in the previous version) and also the Discussion section (1022 words versus 1505 in the previous version).

All changes appear as track changes to better locate them in the text.

Reviewer 2 Report

The author shows the cross-sectional retrospective study of non-splenectomized and never treated patients with type 1GD.

Basically, the research design is appropriate and the methods and the results are well described.

It's good enough to be published.

Minor comments:

1   page 6, line 15. 12/16 is not 81.2%, but 75%.

2   page 7, The author describes five bone complications in the text. But, they are different from the patients' symptoms in Table 4. In addition, 6 patients in Table 4 and 8 patients in Table 5 are similar and it's really confusing, so they should combine them into one table.

Author Response

1   page 6, line 15. 12/16 is not 81.2%, but 75%.

This sentence has been deleted according to reviewer’s comments

2   page 7, The author describes five bone complications in the text. But, they are different from the patients' symptoms in Table 4. In addition, 6 patients in Table 4 and 8 patients in Table 5 are similar and it's really confusing, so they should combine them into one table

The description of bone complications has been homogenized, and Tables 4 and 5 have been combined.

Round 2

Reviewer 1 Report

Thank you for updating the manuscript. It still feels too voluminous for the meritorical content, however it can be presented to a wider audience in the current form.